# Blue Light Enhances the Antioxidant, Antimicrobial, and Antitumor Potential of the Green Microalgae *Coelastrella* sp. BGV

**DOI:** 10.3390/plants13233295

**Published:** 2024-11-23

**Authors:** Zhaneta Georgieva, Zornitsa Karcheva, Tanya Toshkova-Yotova, Ani Georgieva, Reneta Toshkova, Detelina Petrova, Miroslava Zhiponova, Ganka Chaneva

**Affiliations:** 1Department of Plant Physiology, Faculty of Biology, Sofia University “St. Kliment Ohridski”, 1164 Sofia, Bulgaria; zkarcheva@uni-sofia.bg (Z.K.); detelina@biofac.uni-sofia.bg (D.P.); 2Laboratory “Experimental and Applied Algology”, Institute of Plant Physiology and Genetics, Bulgarian Academy of Sciences, 1113 Sofia, Bulgaria; t_toshkova_yotova@abv.bg; 3Institute of Experimental Morphology, Pathology and Anthropology with Museum, Bulgarian Academy of Sciences, 1113 Sofia, Bulgaria; georgieva_any@abv.bg (A.G.); reneta.toshkova@gmail.com (R.T.)

**Keywords:** antimicrobial activity, antioxidant activity, antitumor activity, *Coelastrella*, green microalgae, LED

## Abstract

Green algae of the genus *Coelastrella* have attracted the attention of scientists due to their rich biochemical composition and potential for application in phytomedicine. The present study investigated the influence of light on the bioactive capacity of extracts from the Bulgarian strain of the green microalgae *Coelastrella* sp. BGV. Three LED lights were examined—red/blue (C1), blue (C2), and control white light (C3). The respective ethanol extracts were analyzed for the total content of phenolic antioxidants. The antimicrobial activity was tested using the disk-diffusion method against 10 microorganisms. The antiproliferative and cytotoxic effects on cervical carcinoma HeLa and hepatocellular carcinoma HepG2 cell lines, as well as non-tumorigenic embryonal fibroblasts BALB/3T3 control, were evaluated using a cell viability assay. The overall results highlighted blue light as a factor enhancing the antioxidant, antibacterial, and cytotoxic activities of the C2 microalgal extract. Additionally, the investigated mechanism of the antitumor activity revealed a proapoptotic effect. In contrast, the C1 extract exhibited weaker activity and selectivity, while the C3 extract was the least active but demonstrated high cytotoxic selectivity. This study could contribute to expanding knowledge about the high biological potential of green microalgae and the development of biotechnological approaches for its regulation.

## 1. Introduction

Microalgae represent the most efficient biological system for absorbing solar energy and producing organic compounds through photosynthesis. They have been studied as green energy generators [1]; important food sources for humans and animals [2]; unparalleled feed for aquaculture [3]. Microalgae are a promising source of pharmacologically active compounds with potential therapeutic applications, showing anticancer, anti-inflammatory, antiviral, antifungal, and antibacterial properties [4,5,6]. Extensive research on marine microalgae has led to the identification of over 15,000 bioactive molecules, including fatty acids, sterols, phenolic compounds, terpenes, enzymes, polysaccharides, alkaloids, toxins, and carotenoids [7]. Among the natural antimicrobial agents derived from microalgae are fatty acids, indoles, acetogenins, phenols, terpenes, and volatile halogenated hydrocarbons [8]. Notably, many microalgal compounds, particularly free fatty acids, have demonstrated strong antibacterial effects, paving the way for potential drug development targeting human pathogens and applications as food preservatives, with no bacterial resistance to these fatty acids reported to date [9]. More than half of marine microalgae species have shown promise for isolating bioactive agents that can trigger apoptosis in cancer cells [10]. Clinical research suggests that phytomedicine, including algae-derived treatments, may be beneficial in managing conditions like prostatic hyperplasia due to carotenoids and their effects on vascular and cellular growth processes [11]. The activity of these compounds is thought to involve mechanisms such as enzyme modulation, antioxidant protection, immune system enhancement, regulation of hormone and growth factor signaling, and control of cell cycle progression, differentiation, and apoptosis [11,12].

Microalgae also offer an important biotechnological advantage: they possess remarkable metabolic flexibility, allowing them to serve as natural bioreactors for the targeted biosynthesis of health-promoting compounds [13,14]. The production of bioactive substances by microalgae is highly dependent on cultivation conditions, with factors like light, temperature, salt content, nutrient levels, and cultivation duration all influencing the yield and composition of valuable metabolites and related biological activities [15,16,17,18]. Light intensity and its spectrum directly affect the quantity and quality of biomass and the biochemical composition of microalgae to species-specific degrees [15,19,20,21,22]. Similar to bacteria and fungi [23], microalgae can be stimulated for targeted production of bioactive metabolites by modulation of the cultivation conditions and induction of cellular stress [24]. Microalgae have developed several mechanisms to mitigate the harmful impact of reactive oxygen species (ROS), including increased synthesis of non-enzymatic antioxidants such as chlorophylls [25], carotenoids [26], phycobiliproteins [27], and phenolics [28], as well as enzymatic antioxidants [29].

In recent years, representatives of the green algae genus *Coelastrella* have attracted the attention of scientists due to their interesting biochemical composition, which includes chlorophyll *a*, chlorophyll *b*, carotenoids (astaxanthin, lutein, canthaxanthin, and β-carotene), and fatty acids (linoleic acid, oleic acid, and palmitic acid) [30,31,32]. Information is gradually appearing in the literature about the potential of these algae as a source for the production, cultivation, and application of their carotenoids in biomedicine and the use of their lipids for biofuels [31,33,34,35].

The choice of an appropriate light source allows for controlled modification of both quality and quantity of microalgal biomass, optimizing biomass productivity and enhancing the content of valuable metabolites for targeted applications [36]. LED lights have shown high efficiency in promoting the rapid growth of microalgae, with blue and red wavelengths demonstrating the best results in boosting biomass production [37]. The object of investigation in the present scientific study is the promising microalgal strain *Coelastrella* sp. BGV cultivated under different LED lights. The aim is to investigate the influence of the light spectrum on the antioxidant, antibacterial, and antitumor potential of ethanol extracts.

## 2. Results

### 2.1. Pigment Content of Coelastrella sp. BGV

The microalgae *Coelastrella* sp. BGV was cultivated under three different lights as indicated in Figure 1. The highest content of plastid pigments was recorded under blue light (C2), where the amounts of chlorophyll *a* were up to 2.4 times higher than those of the control (C3), and the content of carotenoids increased by approximately 60% (Table 1). The red/blue light (C1) enhanced the levels of chlorophyll *a*, *b* and carotenoids, with values exceeding those of the control by about 56% for chlorophylls and 33% for carotenoids.

### 2.2. Antioxidant Capacity of Coelastrella sp. BGV Ethanol Extracts

The solvent efficiency test showed that the ethanol extracts were significantly more enriched in phenolic antioxidants than aqueous extracts (Appendix A). The yield of ethanol extracts significantly increased under red/blue (C1) and blue lights (C2), indicating enhanced biosynthetic activity compared to the control (C3) with white light (Table 2). To assess the influence of light on the antioxidant potential of *Coelastrella* sp. BGV, we analyzed the ethanol extracts for changes in total phenolic and flavonoid content, as well as total antioxidant activity (TAA) (Table 2).

The highest phenolic and flavonoid content values were achieved in the C2 extract, showing an increase of 30% and 2.7-fold, respectively, compared to the C3 control. The TAA level of the C1 extract increased by 47% compared to the control C3.

### 2.3. Antimicrobial Activity of Coelastrella sp. BGV Ethanol Extracts

The C1, C2, and C3 microalgal extracts were further tested against 5 Gram-negative and 4 Gram-positive bacterial strains, as well as 1 fungal strain *Candida albicans* (Table 3). The C2 extract exhibited the strongest antibacterial activity, inhibiting the growth of 5 out of the 9 tested pathogenic bacteria, while the C1 extract inhibited only 3 bacterial strains (Table 3). The control C3 extract showed no activity, as it did not affect microbial growth. Among the Gram-negative strains, *Enterobacter cloacae* was the most sensitive, while the Gram-positive bacteria *Bacillus cereus* and *Staphylococcus aureus* had their growth suppressed by both C1 and C2. The Gram-positive bacteria *Enterococcus faecalis* and *B. subtilis*, along with the Gram-negative bacteria *Proteus mirabilis* and *Pseudomonas aeruginosa*, remained completely resistant to the studied *Coelastrella* sp. BGV extracts. The growth of the fungal pathogenic strain was not affected by any of the microalgal extracts.

### 2.4. In Vitro Antitumor Activity of Coelastrella sp. BGV Ethanol Extracts

The in vitro antiproliferative/antitumor activity of the C1, C2, and C3 ethanol extracts of *Coelastrella* sp. BGV was evaluated against human cancer cell lines HeLa and HepG2, as well as non-cancerous BALB/3T3 mouse fibroblasts, using the MTT assay. Cell cultures from the studied cell lines were exposed to increasing concentrations (31.3, 62.5, 125, 250, and 500 µg/mL) of microalgal extracts (C1, C2, and C3) for 24 and 48 h, when cell viability was determined.

#### 2.4.1. Cytotoxic Effect of Microalgal Extracts Against HeLa Cervical Tumor Cells

A statistically significant, concentration- and time-dependent decrease in the cell viability of HeLa tumor cells treated with the C2 extract was observed at all tested concentrations for both time intervals, with a strong effect noted at 48 h (C2, Figure 2). The cell viability values determined at the 48th h were 2 to 5 times lower than those at the 24th h and fell below 50% for all tested concentrations. A similar trend, though with a slightly weaker inhibitory effect, was observed for the C1 extract (Figure 2). After treatment with C1 and C2 extracts at a concentration of 500 μg/mL, cell viability was 0% at both time points tested. The weakest antitumor activity was observed with the C3 control extract (Figure 2). In this case, the percentage of viable cells was significantly below 50% at the 24th and 48th h at concentrations of 250 μg/mL and 500 μg/mL (Figure 2).

#### 2.4.2. Cytotoxic Effect of the Microalgal Extracts Against HepG2 Hepatocellular Tumor Cells

The strongest impact was observed in the case of C2 ethanol extract on HepG2 tumor cells at both time intervals, with the inhibition being more pronounced at 48 h (Figure 3). The C1 ethanol extract decreased the cell proliferation significantly, but to a much lesser extent than the C2 extract (Figure 3). The weakest antiproliferative effect in HepG2 was exhibited by the C3 extract. Cell viability was reduced below 50% at a concentration of 500 μg/mL at the 24th hour and at 250 μg/mL at the 48th hour (Figure 3).

#### 2.4.3. Cytotoxic Effect of the Microalgal Extracts Against BALB/3T3 Non-Tumorigenic Control Cell Line

All ethanol extracts of *Coelastrella* sp. BGVs (C1, C2, C3) significantly reduced the viability of the non-tumorigenic BALB/3T3 mouse fibroblasts at 24 and 48 h. Nevertheless, the cell viability values of BALB/3T3 fibroblasts were higher than those of the two tumor cell lines HeLa and HepG2 (Figure 4). The C1 extract induced a significant inhibition of cell proliferation at the 24th h, with BALB/3T3 cell viability reduced below 50% at a concentration of 125 μg/mL (Figure 4). Higher concentrations induced 100% cytotoxicity. At the 48th hour, all tested concentrations decreased cell viability below 50%, reaching 0%. A similar trend was observed with the C2 extract, but the inhibitory effect was less pronounced (Figure 4). The viability of BALB/3T3 mouse fibroblasts was least affected after treatment with the C3 extract, where no complete inhibition was observed (Figure 4).

#### 2.4.4. Determination of IC_50_ Concentration and Selectivity Index

The half maximal inhibitory concentration (IC_50_), defined as the concentration required to reduce cell viability by 50%, was calculated for the ethanol extracts of *Coelastrella* sp. BGV (Table 4). In treatments of HeLa cells, the C2 extract exhibited the highest cytotoxicity, followed by the C1 extract, with the C3 extract showing the least cytotoxic effect. A similar trend was observed in HepG2 cells.

In contrast, the BALB/3T3 non-tumorigenic cells showed better survival rates after treatment with the C3 and C2 extracts at 24 h, and by 48 h, cell viability values across the cell lines became more similar. Among the extracts, C3 was consistently the least toxic, while C1 exhibited the strongest cytotoxic effect.

In addition, the selectivity index (SI) values, which indicate the extracts’ specific (selective) action against tumor cells, were determined. Data are presented in parentheses in Table 4. The highest SI was observed in HeLa cells at the 24th h for the C3 extract. The C2 extract also showed moderate selectivity, which was slightly higher in HepG2 cells, while the C1 extract demonstrated a relatively non-selective action.

#### 2.4.5. Cytomorphological Analysis of Apoptotic Alterations Induced by Coelastrella sp. BGV Extract in HepG2 Cancer Cells

In this study, we aimed to explore the effect of blue light, alone or in combination with red light, on the antitumor activity of microalgae. Given that the C2 extract exhibited the strongest antiproliferative activity and best selectivity against HepG2 human tumor hepatoma cells, we proceeded to investigate its mechanism of action. The HepG2 cells were cultured for 24 h with the microalgal ethanol extract administered at a concentration equal to its IC_50_, as determined by the MTT assay at the 24th h (~60 µg/mL, Table 4). To assess whether the antitumor effect of the microalgal extracts involved apoptosis induction, we performed fluorescence staining to evaluate morphological changes (Figure 5).

In untreated control HepG2 tumor cells, we observed monolayer growth with predominantly round-shaped cells, uniformly green-stained, containing one or more bright green nucleoli and light green to orange-stained intracytoplasmic structures (Figure 5a). In contrast, HepG2 cells cultured with the C2 extract showed severe morphological changes, including disrupted and sparse monolayer growth, pronounced cell polymorphism, an obscured nuclear outline, and the presence of highly rounded or swollen cells with bright orange-red nuclear fluorescence. Chromatin condensation and fragmentation within the nucleus, indicative of late apoptosis, were also observed (Figure 5b). These morphological changes suggest that C2 extract induces cell death via the apoptosis pathway.

To visualize the nuclear morphology, we stained the cells with the fluorescent dye DAPI, which penetrates the cytoplasmic membrane of both living and fixed cells (Figure 5c,d). DAPI staining enabled us to observe DNA-level apoptotic changes, such as chromatin condensation, nuclear fragmentation, and apoptotic body formation. Control, untreated HepG2 cells displayed intact nuclei, slightly oval in shape and variable in size, with smooth margins, homogeneous staining, and evenly distributed chromatin. Cells in various stages of division were also observed (Figure 5c). In cells treated with the C2 extract, damaged nuclear morphology was apparent. The number of nuclei was reduced, and their size and shape showed significant variations. The nuclear contour was irregular, and the chromatin was condensed and unevenly distributed. Cells with nuclear fragmentation and apoptotic body formation were also detected (Figure 5d).

### 2.5. Principal Component Analysis

To correlate the pigments and phenolic compounds of *Coelastrella* sp. BGV with its bioactive potential, data on metabolites and biological activities were analyzed using principal component analysis (PCA) (Figure 6).

Principal component analysis is a powerful tool for visualizing differences among variants based on the complete experimental dataset. The total variance in the data was explained by the first and second principal components (PCs) combined (74.7% + 25.3%). Each experimental parameter contributes to each PC to a different extent, as indicated by the length and orientation of each vector in the PC1/PC2 coordinate system. The closer the vector is to one of the axes, the greater its contribution to that respective PC. Most parameters primarily influence PC1, except for TAA and selectivity against HepG2 cells, which determine PC2, and yield and anti-*E. coli*, *K. pneumoniae* activities, which contribute equally to both PCs. Differences between white and colored lights are reflected along PC1, while the variance between blue and red/blue lights is shown along PC2.

The PCA clearly demonstrates that blue light acts as a factor, stimulating the accumulation of plastid pigments, phenols, and flavonoids. Under blue light, the resulting extracts showed enhanced antibacterial activity, particularly against *E. cloacae*, *S. aureus*, *B. cereus*, *E. coli*, and *K. pneumoniae*. Furthermore, blue light-treated microalgal extracts exhibited strong antitumor activity in HeLa and HepG2 cells, with a marked selectivity against HepG2 cells. Conversely, extracts from cultures grown under white light showed the highest selectivity against HeLa cells. The red/blue light led to increased extract yield and antioxidant activity. Overall, PCA revealed a strong positive correlation between metabolite content (plastid pigments, phenols, and flavonoids) and the antibacterial and antitumor activities of ethanol extracts from *Coelastrella* sp. BGV, pointing the blue light as the optimal cultivation condition for enhanced bioactivity.

## 3. Discussion

In this study, we demonstrated that blue light is a stimulating factor influencing the growth and bioactive potential of the green microalgae *Coelastrella* sp. BGV. Blue light acted as a signal, triggering the accumulation of chlorophylls, carotenoids, and phenolics. These metabolites showed a strong correlation with enhanced antibacterial (against Gram-negative *E. cloacae*, *E. coli*, *K. pneumoniae*, and Gram-negative *B. cereus*, *S. aureus*) and antitumor activities against the HeLa and HepG2 tumor cells. The antibacterial activity exhibited more specific inhibition of Gram-negative bacteria and selectively affected HepG2 tumor cell proliferation by inducing an apoptotic mechanism. Exposure to red light increased the ethanol extract’s yield and antioxidant activity, though other bioactive parameters were not superior to those achieved with blue light alone.

*Coelastrella* sp. BGV is a strain of *Coelastrella* noted for its high growth potential under laboratory conditions and significant carotenoid productivity [39,40]. During the exponential growth phase, its biomass composition was shown to include 35.32% proteins, 38.66% carbohydrates, and 13.9% lipids. In the stationary growth phase, lipid accumulation increased, reaching 21.6% DW [40]. Among the cultivation conditions, temperature and light intensity were highlighted as key factors influencing metabolic productivity, particularly for lipid and carotenoid production [39].

Microalgae-derived chemicals are favored over plant-derived ones due to their diversity in phenolic classes and higher levels of carotenoids and chlorophylls compared to certain plants [41,42]. In this study, we examined the pigment and phenolic content of *Coelastrella* sp. BGV. Blue and red/blue light exposure significantly increased the chlorophyll *a* content, nearly threefold, and boosted carotenoid levels by up to 60%. To increase the concentration of bioactive phytochemicals, the proper selection of an extraction solvent is of utmost importance, and the extraction approach also plays a significant role [43]. Solvents such as ethanol, ethyl acetate, chloroform, and hexane have been shown to achieve higher extraction rates. Water is the most preferable solvent due to its safety. In the current study, we demonstrated that ethanol extracts are more enriched in phenolics compared to aqueous extracts. Ethanol dissolves higher amounts of lipids and is considered a green, safe, and environmentally friendly solvent. It holds GRAS (generally recognized as safe) status from the FDA (United States Food and Drug Administration) and is also approved by EFSA (European Food Safety Authority) [44]. Ethanol extracts of *Coelastrella* sp. BGV have been shown to contain phenolics as follows: 279.0 mg/g DW phenols, 84.3 mg/g DW flavonoids, and 238.5 mM/g DW total antioxidant activity (TAA) [40]. Under our experimental conditions, phenolic compound levels decreased to 43.08 mg/g DW phenols and 8.55 mg/g DW flavonoids, though TAA increased to 329.60 mM/g DW. These findings underscore the importance of cultivation conditions for optimizing microalgal productivity. Both blue and red/blue lights substantially increased phenolic content, particularly flavonoids, as well as TAA.

Natural chlorophylls and phenolics play essential roles in microalgal interactions with the environment. Chlorophylls are crucial for light energy absorption, while phenolic compounds provide UV protection and defense against environmental threats [45]. The heterocyclic structure of these compounds enables them to act as photosensitizers; when combined with visible light and oxygen, they generate reactive oxygen species (ROS) [45,46,47]. This photosensitization is the basis of photodynamic therapy (PDT) [48], where ROS cytotoxicity induces oxidative damage to tumor cells, leading to apoptosis, necrosis, and autophagy. Following the same principle, antimicrobial PDT (aPDT) has been developed, and pigment and phenolic derivatives have been successfully used as antimicrobial agents against infections caused by bacteria, fungi, and viruses, including *S. aureus* and *E. coli* [45,46]. Unlike synthetic compounds, natural products benefit from the evolutionary complexity of their chemical structures, enabling them to participate in physiological processes while also providing defense against abiotic and biotic factors [45].

Ethanol extracts of *Coelastrella* sp. BGV have demonstrated high antimicrobial potential [40]. Growth conditions appear to be crucial, as in our study, extracts from *Coelastrella* sp. BGV cultivated under white LED light showed no antimicrobial activity. However, cultivation under blue LED light restored antibacterial effects against *E. coli* UPEC, *K. pneumoniae*, and *B. cereus*, although no antifungal activity was observed. In a previous study, the fatty acid fraction from *Coelastrella* sp. BGV was found to be active against *E. coli* UPEC, *P. aeruginosa*, *K. pneumoniae*, *B. cereus*, and *C. albicans*, with the highest antibacterial activity observed against *K. pneumoniae* [49]. Supporting this, known antibacterial compounds from microalgae in extreme environments include both saturated and unsaturated fatty acids, along with volatile compounds [50]. Tosheva-Yotova et al. (2022) [49] identified the fatty acid profile in *Coelastrella* sp. BGV, highlighting oleic, linoleic, and palmitic acids as the most abundant in the microalgal biomass. Fatty acids are believed to disrupt bacterial cell membranes, interfering with physiological processes or inducing direct lysis of bacterial cells [51].

Antiproliferative and proapoptotic activities against HeLa tumor cells have been demonstrated for the culture medium, aqueous extract, and sunflower oil extract of *Coelastrella* sp. BGV [52,53]. Tosheva-Yotova et al. (2022) [49] further confirmed the antitumor activity of the fatty acid fraction from *Coelastrella* sp. BGV. The antiproliferative effects at the 24 h treatment mark with the highest tested concentrations of algal extracts were as follows: culture medium (60–72% cell viability at 1000 µg/mL), aqueous extract (66–71% cell viability at 1000 µg/mL), sunflower oil (72–73% cell viability at 1000 µg/mL), and fatty acid fraction (0% cell viability at 400 µg/mL). In comparison, the ethanol extracts tested in this study showed significant inhibition of HeLa cell viability: the control extract (C3) resulted in approximately 40% cell viability at 250 µg/mL, while extracts C1 and C2 reduced cell viability to nearly 0% at 250 µg/mL. The C2 extract showed maximal inhibition of HepG2 tumor cells at 250 µg/mL.

In our study we used as a control normal fibroblast cells that were also affected, with ~0% cell viability at 500 µg/mL for C1 and C2 extracts and ~80% cell viability for the C3 extract at the same concentration. These findings emphasize the importance of dosage and careful application of the extracts enriched in metabolites. Our data revealed an IC_50_ of 63.10 µg/mL for the C2 extract against HepG2 tumor cells, with selective action compared to control non-tumor cells. Previous studies showed that the fatty acid fraction had a much lower IC_50_ of 3.046 µg/mL at 24 h exposure, comparable to the IC_50_ of 2.254 µg/mL for the positive control, doxorubicin [49]. The high antitumor activity of *Coelastrella* sp. BGV fatty acids could be attributed to their high content of palmitic, oleic, and linoleic acids, which have been documented to influence cellular signaling pathways, reactive oxygen species, mitochondrial and cell membrane integrity, and to induce apoptosis [54].

The present study demonstrated the importance of the light spectrum for the microalgal bioactive arsenal. Supporting this, research on the green alga *Scenedesmus obliquus* has shown that blue light induces greater biomass productivity and fatty acid content compared to other LED conditions (red light and red/blue light) [55]. Under red light, *S. obliquus* exhibited higher antioxidant capacity, while under red/blue light (60:40), it produced more carotenoids, particularly lutein and β-carotene, compared to fluorescent white light control. Schuelter et al. [56] studied the impact of different light sources on the antibacterial activity of extracts from newly isolated green microalgae, finding that cultivation under green, red, and blue LED light increased antibacterial activity compared to white LED or fluorescent light [56].

## 4. Materials and Methods

### 4.1. Microalgal Cultivation

The microalgae *Coelastrella* sp. strain BGV belongs to the Collection of the Laboratory of Experimental Algology, Institute of Plant Physiology and Genetics, BAS [39,40]. For the experiment, the algal suspension was cultivated in Šetlik medium modified by Georgiev et al. [57] at 28–30 °C, under continuous LED illumination. Carbon nutrition was provided by bubbling 2% CO_2_ (*v*/*v*) in air (100 L/h). The strain was intensively cultivated at a laboratory setup, under three LED lights: red/blue (blue 460 nm:red 660 nm:far red 730 nm, in a ratio of 15:75:10%); blue (royal blue, 460 nm); and white light as a control (OSRAM Opto Semiconductors, Brno, Czech Republic, 150 μmol phot m^−2^ s^−1^). The light intensity and spectrum were recorded using a SpectraPen mini (Photon Systems Instruments/PSI/, Brno, Czech Republic). After 7 days of cultivation, the biomass was harvested and used for the preparation of algae extracts and subsequent analyses.

### 4.2. Preparation of Algal Extracts

Ethanol extraction was performed according to Toshkova-Yotova et al. [40]. Briefly, lyophilized biomass was incubated in 96% ethanol (1:10 ratio) while the mixture was heated to 50 °C for 4 h. The extract was filtered and evaporated to dry residue at 60 °C.

### 4.3. Pigments

The amount of chlorophyll *a*,*b* and carotenoids in the algal biomass was determined spectrophotometrically at wavelengths of 665 nm (chlorophyll *a*), 645 nm (chlorophyll *b*), and 460 nm (carotenoids) after extraction with boiling methanol. The pigments’ content was calculated according to the formulas of McKinney (1941) [58]:

Chlorophyll *a*:$$\%ADW=\frac{\left[\left(0.0127.E665nm\right)-\left(0.00269.E645nm\right)\right].Vf.100}{V0.ADW}$$

Chlorophyll *b*:$$\%ADW=\frac{\left[\left(0.0229.E645nm\right)-\left(0.00468.E665nm\right)\right].Vf.100}{V0.ADW}$$

Carotenoids:$$\%ADW=\frac{\left[E460nm-\left(3.917.A+130.3.B\right)\right].Vf.100}{V0.ADW}$$
A=[(0.0127.E665nm)−(0.00269.E645nm)]
B = [(0.0229.E645 nm) − (0.00468.E665 nm)]
where Vo—initial volume of the algal suspension (mL); Vf—final volume of the extract (mL); ADW—absolutely dry weight (mg/mL)

### 4.4. Antioxidant Activity

The amount of total phenols was estimated by the Folin–Ciocalteu method [59], the flavonoids by the AlCl_3_ colorimetric method [60], and the total antioxidant activity (TAA) was determined according to [61]. The quantification of total phenolics and antioxidant activities was based on standard curves created using the respective standards: gallic acid (GA) for phenolics, quercetin (Q) for flavonoids, and α-Tocopherol (α-TF) for TAA. The calculations were performed according to the following formulas:$$\mathrm{Total}\;\mathrm{phenolics}\;(\mathrm{mg}/g\;\mathrm{DW})=\frac{\left(C*V\right)}{\left(DW*1000\right)}$$
where C—standard (GA or Q) quantity (mg/mL); V—extract volume (mL); DW—dry weight (g); 1000—coefficient for conversion from µg in mg.
$$\mathrm{Total}\;\mathrm{antioxidant}\;\mathrm{activity}\;(\mathrm{mM}\;\alpha -\mathrm{TF}/g\;\mathrm{DW})=\frac{\left(C*V\right)}{\left(DW*1000\right)}$$
where C—α-TF standard quantity (µM/mL); V—extract volume (mL); DW—dry weight (g); 1000—coefficient for conversion from µM in mM.

### 4.5. Antimicrobial Activity

The antimicrobial efficiency of the *Coelastrella* sp. BGV ethanol extracts was tested against 9 bacterial strains: Gram-positive bacteria: *Enterococcus faecalis* NBIMCC 3360, *Bacillus subtilis* NBIMCC 1709, *Bacillus cereus* NBIMCC 1085, *Staphylococcus aureus* ATCC 25923; Gram-negative bacteria: *Enterobacter cloacae* NBIMCC 8570, *Escherichia coli* NBIMCC 8954 (UPEC), *Proteus mirabilis* NBIMCC 8747, *Pseudomonas aeruginosa* NBIMCC 3700, *Klebsiella pneumoniae* NBIMCC 3670; and 1 fungal strain—*Candida albicans* NBIMCC 74. The strains were obtained from NBIMCC (National Bank of Industrial Microorganisms and Cell Cultures), Bulgarian Type Culture Collection. The experiments were performed by the disk-diffusion method Essawi [62].

Dried algal residues were dissolved in 10% *v*/*v* DMSO in sterile distilled water up to a concentration of 100 mg/mL and sterilized by filtration through a 0.45 μm filter. Sterile filter paper discs were impregnated with 25 μL extracts of each sample. As a positive control, ciprofloxacin disks (5 μg/disc) were used for bacterial strains, and nystatin disks (50 µg/mL) for the fungal pathogenic strain. Growth inhibition zones were measured after 24 h of incubation at 37 °C.

### 4.6. Antitumour Activity

#### 4.6.1. Cell Cultures

Cell cultures from permanent human tumor cell lines—HeLa (CCL-2; human cervical carcinoma), HepG2 (HB-8065, human hepatocellular carcinoma), and non-tumor mouse embryonic fibroblasts—BALB/3T3 (CCL-163; control cell line) were used as model systems in the in vitro experiments. The cell lines were purchased from the American Type Culture Collection (ATCC, Manassas, VA, USA).

Cells were cultured in Dulbecco’s modified Eagle’s medium (DMEM) (Sigma-Aldrich, Germany) enriched with 10% heat-inactivated fetal bovine serum (Gibco, Austria), 2 mM L-glutamine, 100 U/mL penicillin, and 0.1 mg/mL streptomycin in 50 cm^2^ and 75 cm^2^ plastic tissue culture vessels. The cultures were maintained in a logarithmic growth phase at 37 °C, 95% humid air with 5% CO_2_ content. For in vitro experiments, cells were dissociated from the bottom of the culture vessels by trypsinization with 0.25% Trypsin-EDTA (Sigma-Aldrich, Germany), washed with phosphate-buffered saline (PBS pH 7.4), adjusted to the desired concentration, and seeded in 96- or 24-well tissue culture plates depending on the test method. After 24 h of cultivation in a CO_2_ incubator (3111, Thermo Scientific, Waltham, Massachusetts, USA) to obtain a monolayer, the cells were treated with rising concentrations of microalgal extracts (31.25 to 500 µg/mL) for the MTT cell viability assay or were exposed for 24 h at a dose equal to the IC_50_ determined by the MTT test at the 24th h for the cytomorphological assay.

#### 4.6.2. Cell Proliferation/Viability Assay

Cell proliferation was assessed by the MTT (3-(4,5-dimethylthiazol-2-yl)-2,5-diphenyl tetrazolium bromide) test according to the approach of Mossmann [63]. The spectrophotometric method is based on the reduction of the yellow tetrazolium salt MTT to violet formazan crystals by living cells. The formation of formazan crystals is proportional to the activity of mitochondrial enzymes and, accordingly, to cell viability.

Cells at a concentration of 1 × 10^4^ cells/well in DMEM medium with 10% FBS (fetal bovine serum) were seeded in 96-well tissue culture plates at a volume of 100 µL/well. After 24 h of incubation in a CO_2_ thermostat for good cell adhesion and spread, the medium was replaced with 100 µL of fresh medium containing a series of concentrations of each microalgal extract (31.25, 62.5, 125, 250, and 500 µg/mL), and the cells were reincubated for 24 and 48 h under the same conditions. Untreated cells were used as a negative control. At the end of the 24 h and 48 h, the culture medium was removed and 100 µL of MTT solution (concentration 0.5 mg/mL) was added to each well. After additional incubation for 3 h in a CO_2_ thermostat, the formazan crystals were dissolved in 100 µL/well lysis solution (DMSO:ethanol = 1:1). The optical density (OD) of the dissolved formazan was measured at 570 nm and 620 nm (as reference length) on an ELISA spectrophotometer (TECAN, SunriseTM, Grödig/Salzburg, Austria). Cell viability was calculated by the formula:CV(%) = OD_570_(experiment sample)/OD_570_(control sample) × 100.

#### 4.6.3. Determination of Half Maximal Inhibitory Concentration and Selectivity Index

The half maximal inhibitory concentration (IC_50_) (µg/mL) of each microalgal extract was calculated as the concentration that caused 50% inhibition of cell proliferation. To determine the cytotoxic selectivity of the microalgal extracts towards the used tumor cell lines, the selectivity index (SI) was calculated according to the following equation:SI = IC_50_ normal cells/IC_50_ tumor cells.

### 4.7. Evaluation of the Morphological Alterations by Fluorescent Tests

#### 4.7.1. Acridine Orange and Ethidium Bromide Fluorescent Staining

Cytomorphological changes of tumor cells cultured in the presence of microalgal extracts were studied after the application of double intravital staining with fluorescent dyes acridine orange (AO) and ethidium bromide (EtBr) according to standard procedure Abdel [64]. For this purpose, tumor cells at a concentration of 1 × 10^5^ cells/mL were instilled on glass lamellas placed on the bottom of each well in a 24-well plate and incubated overnight in a CO_2_ incubator to form a monolayer on its surface.

Next, the culture medium was replaced with a new medium containing microalgal extracts at a concentration equal to the IC_50_ determined by the MTT assay. Tumor cells cultured on glass lamellae only in the nutrient medium were used as a control. After an additional 24 h incubation, the lamellae were washed with PBS, drained on filter paper, and placed on slides with a mixture of fluorescent dyes (equal amounts of 10 µL/mL AO and 10 µL/mL EtBr). The morphology changes of freshly stained tumor cells were observed and photographed within 5–10 min under staining with a fluorescence microscope (Leica DM 5000B, Wetzlar, Germany) before the fluorescent color began to fade.

#### 4.7.2. DAPI Staining

Staining with DAPI (4′, 6-diamidino-2-phenylindole) was used to determine the nuclear morphology of treated and untreated tumor cells. Tumor cells were cultured and treated with microalgal extracts for 24 h, as described for the AO/EtBr staining. After incubation, the glass lamellae with adherent cells were washed twice with PBS, fixed with 3% paraformaldehyde for 10 min at room temperature, and stained with DAPI solution for 20 min at room temperature in the dark (according to the protocol and manufacturer’s instructions). Samples with stained cells were coated with Mowiol^®^ and mounted on slides. The nuclear morphology of the treated and control tumor cells was examined under a fluorescence microscope (Leica DM 5000B, Wetzlar, Germany)**.**

### 4.8. Statistical Analysis

All experiments were performed at least in triplicate. The results are presented as mean values ± standard deviations (±SD) or as a representative mean value. One-way ANOVA, followed by the Holm–Sidak test, was performed for statistical comparison between the experimental variants. Different letters denote significant variations (*p* ≤ 0.05). In the case of the treatments with a series of concentrations of microalgal ethanol extracts, the results are presented as the means ±SD. Here, the significance of the differences between the control and experimental groups was determined by one-way analysis of variance (ANOVA) followed by Bonferroni’s test using GraphPad PRISM software, Version 5 (GraphPad Software Inc., San Diego, CA, USA). The values of *p* < 0.05 were considered statistically significant compared to the untreated control and indicated by asterisks. The inhibitory concentration (IC_50_) (µg/mL) of each microalgal extract was calculated using non-linear regression curve fit analysis (GraphPad PRISM). The Selective Index (SI) value was calculated from the IC_50_ ratio between normal cells versus the IC_50_ of cancer cells. Principal component analysis (PCA) of parameters of the examined extract variants was performed by the *prcomp* function from the *stats* package in R 4.4.1 programming language with centering to zero and scaling to unit variance of the experimental variables. PCA graphs were plotted by the R package *ggbiplot* 0.55.

## 5. Conclusions

The present results demonstrated that the biological activity of the ethanol extracts obtained from *Coelastrella* sp. BGV can be modulated by varying the light illumination conditions used during the microalgal cultivation. Comparing the effects of lights with different spectral characteristics, the blue light acted as the strongest stimulator of the antioxidant, antimicrobial, and antiproliferative activities of the microalgal extracts. Our study proves that ethanol extracts of *Coelastrella* sp. BGV cultured under blue LED illumination have the potential as a therapeutic and chemopreventive agent for cancer treatment. Although ethanol is known as a safe solvent, future perspectives could include testing alternative solvents, such as the green solvents, natural deep eutectic solvents (NADES), which have been proposed for the efficient extraction of a wide range of natural compounds, including polyphenols [65].

## Figures and Tables

**Figure 1 plants-13-03295-f001:**
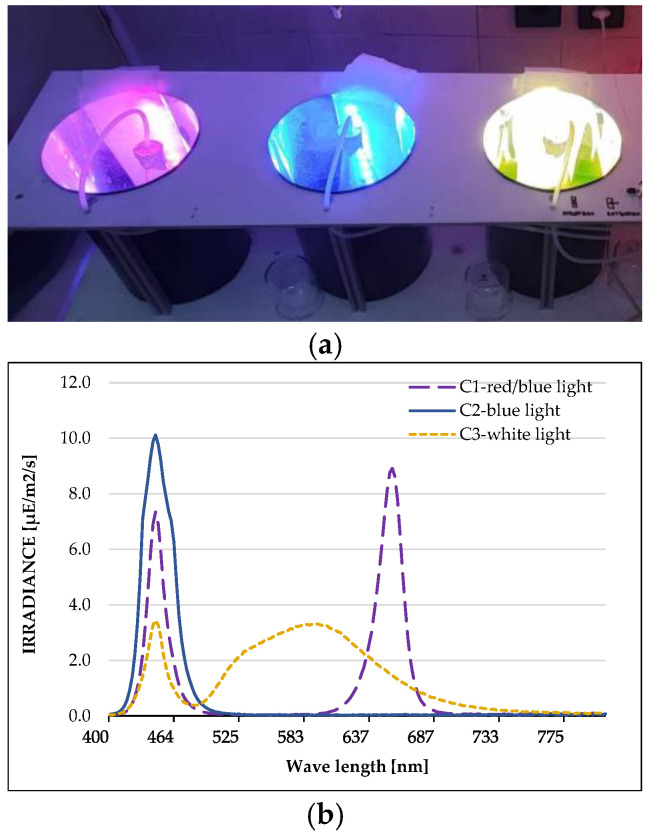
Lights used for cultivation of *Coelastrella* sp. BGV. (**a**) Experimental system with LED lights. (**b**) Spectrum of each light: C1—red/blue light; C2—blue light; C3—white light.

**Figure 2 plants-13-03295-f002:**
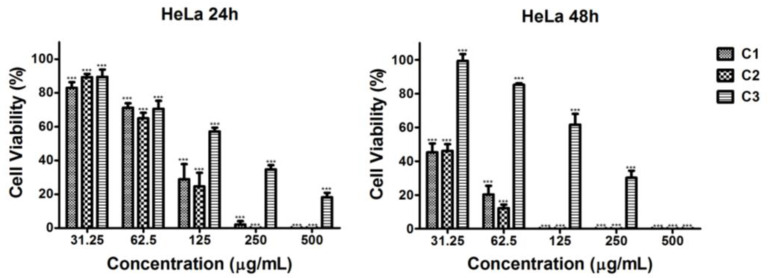
Effect of series of concentrations of ethanol extracts of *Coelastrella* sp. BGV cultivated at different illumination conditions (C1—red/blue light; C2—blue light; C3—white light) on the proliferation of HeLa cells, determined at 24th h and 48th h by MTT test. Data are presented as mean ± SD (*n* = 6); *** *p* < 0.001 is a statistical difference from the untreated control (i.e., 100% cell viability).

**Figure 3 plants-13-03295-f003:**
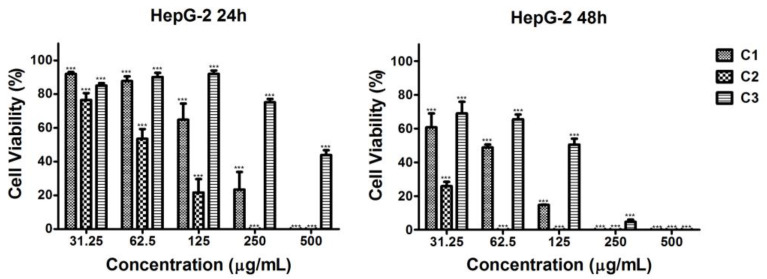
Effect of series of concentrations of ethanol extracts of *Coelastrella* sp. BGV cultivated at different illumination conditions (C1—red/blue light; C2—blue light; C3—white light) on the proliferation of HepG2 tumor cells, determined at 24th h and 48th h by MTT test. Data are presented as mean ± SD (*n* = 6); *** *p* < 0.001 is a statistical difference from the untreated control (i.e., 100% cell viability).

**Figure 4 plants-13-03295-f004:**
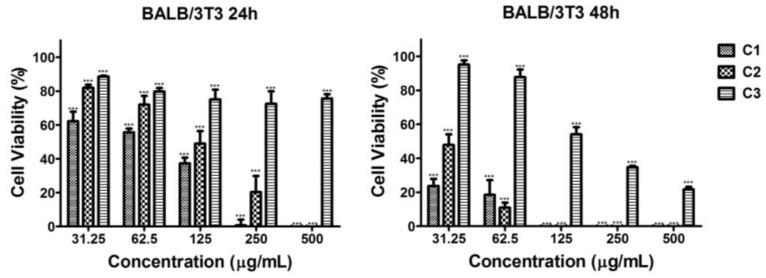
Effect of series of concentrations of ethanol extracts of *Coelastrella* sp. BGV cultivated at different illumination conditions (C1—red/blue light; C2—blue light; C3—white light) on the proliferation of BALB/3T3 mouse fibroblasts (non-tumor cells), determined at 24th h and 48th h by MTT test. Data are presented as mean ± SD (*n* = 6); *** *p* < 0.001 is a statistical difference from the untreated control (i.e., 100% cell viability).

**Figure 5 plants-13-03295-f005:**
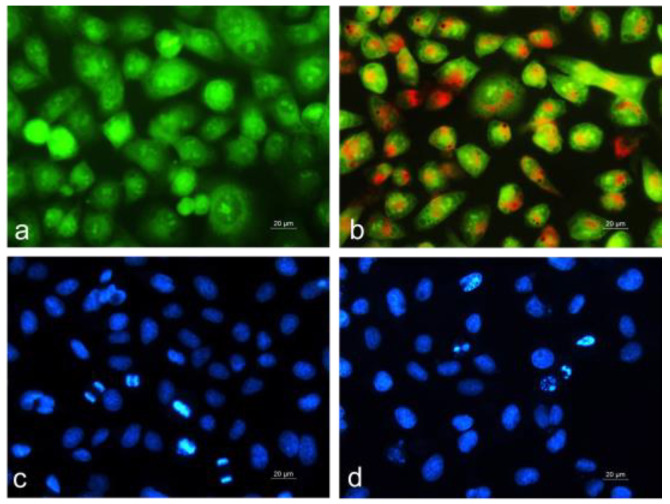
Fluorescence microscopy of human HepG2 tumor cells after treatment with ethanol extract of *Coelastrella* sp. BGV cultivated under blue light (C2) at IC_50_~60 µg/mL. (**a**,**c**) untreated control HepG2 cells; (**b**,**d**) HepG2 cells treated with C2 extract. Upper row, (**a**,**b**), vital staining with acridine orange/ethidium bromide: (**a**)—control cells; (**b**)—cells with a bright green nucleus with chromatin condensation (early apoptosis); late apoptosis—cells with an orange-red nucleus with chromatin condensation. Lower row, (**c**,**d**), DAPI staining of nuclei in blue: (**c**)—control; (**d**)—chromatin condensation, nuclear fragmentation, and formation of apoptotic bodies signs of apoptosis. Scale bar = 20 µm.

**Figure 6 plants-13-03295-f006:**
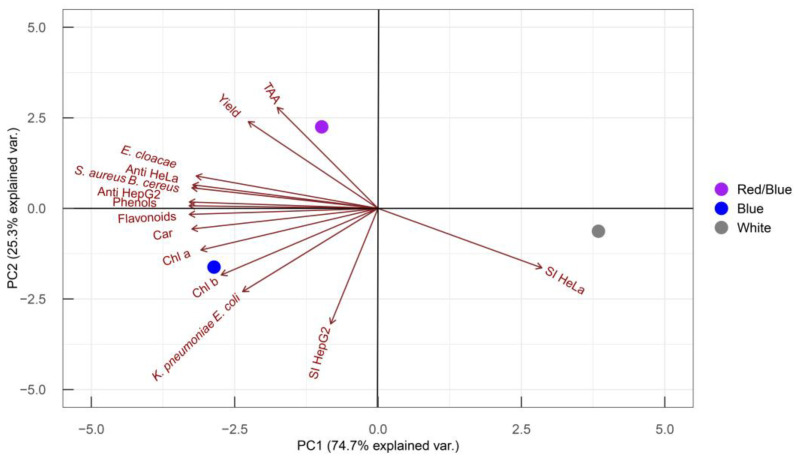
PCA represents differences between cultivation variants of *Coelastrella* sp. BGV (red/blue, blue and white lights) according to the metabolites and the biological activities of ethanol extracts examined in this study (arrows). Overlapping arrows correspond to parameters with high positive correlation; linearly opposing vectors point to highly negatively correlated variables; and orthogonally positioned variables characterize with no correlation between them.

**Table 1 plants-13-03295-t001:** Pigment content of *Coelastrella* sp. BGV cultivated at different illumination conditions (C1—red/blue light; C2—blue light; C3—white light). Data are presented as mean ± SD (*n* ≥ 6). A one-way ANOVA (Holm-Sidak test) was applied to determine statistical differences between the extract variants, which are denoted by different letters for each pigment (*p* ≤ 0.05).

Extracts	Chlorophyll *a*,	Chlorophyll *b*,	Carotenoids,
	%DW	%DW	%DW
C1	0.25 ± 0.004 ^b^	0.02 ± 0.003 ^b^	0.20 ± 0.001 ^b^
C2	0.38 ± 0.002 ^a^	0.04 ± 0.004 ^a^	0.24 ± 0.005 ^a^
C3	0.16 ± 0.004 ^c^	0.02 ± 0.006 ^c^	0.15 ± 0.006 ^c^

**Table 2 plants-13-03295-t002:** Yield, total phenolic and flavonoid content, and TAA of ethanol extracts of *Coelastrella* sp. BGV cultivated at different illumination conditions (C1—red/blue light; C2—blue light; C3—white light). Data are presented as mean ± SD (*n* ≥ 6). A one-way ANOVA (Holm-Sidak test) was applied to determine statistical differences between the extract variants, which are denoted by different letters for each parameter (*p* ≤ 0.05).

Extracts	Yield	Phenols	Flavonoids	TAA
	%DW	mg/g DW	mg/g DW	mM/g DW
C1	27.31 ± 0.197 ^a^	52.56 ± 0.001 ^b^	18.35 ± 0.002 ^b^	484.15 ± 0.003 ^a^
C2	25.00 ± 0.158 ^b^	55.89 ± 0.001 ^a^	23.09 ± 0.006 ^a^	378.39 ± 0.013 ^b^
C3	22.81 ± 0.258 ^c^	43.08 ± 0.001 ^c^	8.55 ± 0.0005 ^c^	329.60 ± 0.002 ^c^

**Table 3 plants-13-03295-t003:** Antimicrobial activity of ethanol extracts of *Coelastrella* sp. BGV cultivated at different illumination conditions (C1—red/blue light; C2—blue light; C3—white light). Data are presented as mean ± SD (*n* ≥ 3). The zone of inhibition showed a bactericidal effect. * Ciprofloxacin (for the bacteria) and nystatin (for the fungus) were positive antibiotic controls.

	Zone of Inhibition (mm)									
Extracts	Gram-Negative Bacterial Strains					Gram-Positive Bacterial Strains				Fungal Strain
	*Enterobacter cloacae*	*Escherichia coli*	*Proteus mirabilis*	*Pseudomonas aeruginosa*	*Klebsiella pneumoniae*	*Enterococcus faecalis*	*Bacillus subtilis*	*Bacillus cereus*	*Staphylococcus aureus*	*Candida albicans*
C1	8 ± 0.5	-	-	-	-	-	-	8 ± 0.5	8 ± 0.5	-
C2	8 ± 0.5	9 ± 0.5	-	-	8 ± 0.5	-	-	9 ± 0.5	9 ± 0.5	-
C3	-	-	-	-	-	-	-	-	-	-
Antibiotic *	28 ± 2	20 ± 1	15 ± 0.5	25 ± 2	30 ± 2	12 ± 0.5	30 ± 2	15 ± 0.5	25 ± 2	15 ± 0.5

**Table 4 plants-13-03295-t004:** IC_50_ * and Selectivity index (SI) ** values of ethanol extracts of *Coelastrella* sp. BGV cultivated at different illumination conditions (C1—red/blue light; C2—blue light; C3—white light).

Extracts	IC_50_ HeLa		IC_50_ HepG2		IC_50_ BALB/3T3	
	24 h	48 h	24 h	48 h	24 h	48 h
C1	84.50 ± 6.68 ^b^ **(0.72)**	29.05 ± 2.45 ^a^ **(0.48)**	153.80 ± 16.35 ^b^ **(0.40)**	48.18 ± 6.34 ^b^ **(0.29)**	61.01 ± 4.47 ^a^	14.01 ± 1.65 ^a^
C2	78.58 ± 7.82 ^a^ **(1.39)**	29.60 ± 1.64 ^a^ **(1.03)**	63.10 ± 9.28 ^a^ **(1.73)**	29.59 ± 0.18 ^a^ **(1.03)**	109.10 ± 9.96 ^b^	30.39 ± 2.29 ^b^
C3	148.80 ± 12.29 ^c^ **(>3.36)**	155.90 ± 15.06 ^b^ **(1.08)**	457.20 ± 33.00 ^c^ **(›1.09)**	85.86 ± 11.79 ^c^ **(1.96)**	>500.00 ^c^	168.30 ± 10.31 ^c^

* IC_50_ (µg/mL) values of the microalgal ethanol extracts on cancer (HeLa; HepG2) and normal (BALB/3T3) cell lines, determined at 24th and 48th. Data are presented as mean ± SD (n = 6). A one-way ANOVA (Holm-Sidak test) was applied to determine statistical differences between the extract variants, which are denoted by different letters for each time point (*p* ≤ 0.05). ** The SI value (in parentheses, in bold) was calculated from the IC_50_ ratio of BALB/3T3 cells versus IC_50_ of cancer cells HeLa or HepG2, respectively. The selectivity score was separated into three categories: SI values ≤ 1—non-selective action; 1 < SI values < 5—moderate selectivity; SI values > 5—selective action [38].

## Data Availability

The data presented in this study are available in this article.

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
