# Peer review of "Blue Light Enhances the Antioxidant, Antimicrobial, and Antitumor Potential of the Green Microalgae Coelastrella sp. BGV"

_plants, 2024, doi:10.3390/plants13233295_

Round 1

Reviewer 1 Report

Comments and Suggestions for Authors

The paper is well-written and shows a progression toward understanding light wavelength on metabolic content and its potential application as an antimicrobial or anticancerous agent.

Considering that the extract with the highest chlorophyll content and phenolic content showed the highest cytotoxicity, it would be useful to combine those results as a hypothesis that these compounds, when over-accumulated or not properly managed, may act as photosensitizers that generate more ROS when exposed to light, potentially causing damage to both cancerous and healthy cells if not properly controlled. 

Author Response

We thank the Reviewer for this constructive suggestion. We completely rewrote the Discussion section and included the hypothesis about the possible action of chlorophylls and phenolics as photosensitizers (Lines 296-390; Lines 327-340).

Reviewer 2 Report

Comments and Suggestions for Authors

The manuscript entitled “Blue light is an elicitor of the antioxidant, antimicrobial, and antitumor potential of extracts from the green microalga Coelastrella sp. BGV” by Georgieva et al. presents a study of different color light on the bioactive capacity of extracts from the Bulgarian strain of the green microalga Coelastrella sp. BGV. In this study the authors focused on impact of various color filtered LED lights on the content of phenolic antioxidants and the estimation of antioxidant, antibacterial, and cytotoxic activites of ethanol extracts from this green microalga.

 After reading the entire manuscript, I believe that it requires a lot more work before it being considered acceptable for publication.

 The authors faced the problem of what to call the light. The light of different colors has received the following names: “three different LED light formulas, different LED light conditions, different light quality, different light condition, light variant, quality of light, differences of light variants, LED light sources, the light illumination conditions, influence of light, different light conditions were tested, influence of light quality”.

“blue light extract, white light extract, red/blue extract” are inappropriate.

 The manuscript lacks adequate discussion. The authors should realized that the purpose of the discussion section is to interpret and describe the significance of results obtained in relation to what was already known about the research problem under consideration, and to explain new understanding that emerged as a result of your research.

 The article was prepared carelessly.

 The language is poor enough that the authors should have a third party review it for style before resubmission.

 Detailed comments:

 The title should be edited because (1), strictly speaking, in plant biology, elicitors are substances (see below) and (2) the light affected the microalgae, not the extracts. Maybe something like that: “Blue light enhances the antioxidant, antimicrobial and antitumor potential of the green microalgae Coelastrella sp. BGV”, or something else.

 Comments about “an elicitor”:

In plant biology, elicitors are generally the low-molecular-weight foreign molecules often associated with plant immune response by activating signal cascade. Elicitors are classified into two types, exogenous and endogenous, of elicitors, which can attach to receptor protein of the plant cell membrane.

An elicitor is a molecule that triggers the hypersensitivity response in the plant. Elicitors are very diverse molecules without any chemical similarity, except that they trigger the hypersensitivity response.

 L. 20. What extracts? Specify ethanol extracts.

L. 27. Are you sure about the word “Unlikely”?

L. 37-38. 53, 56 Correct

I would recommend adding data about the effect of the light spectrum on the production of bioactive substances in microalgae to the introduction.

Table 1. What are the values in the table? Mean and SD, n=? This information should be given.

L. 104-108. Delete.

L. 109. “light quality”. Write more specifically.

L.116. Correlation analysis was not carried out.

Table 3. Give the full names of bacteria and fungi..

Figures 2, 3 and 4. To compare the results (for better visualization and clarity), it is better to combine the columns on a single histogram. You can combine the results (columns) for 24 and 48 hours or for different lightings.

Figures 2, 3 and 4. Number of experiments should be indicated in the legends to the figures and tables. So, data are presented as mean ± SD, n=?

Figure 4. What do the letters a, b and c mean? Specify.

L.259-277. This subsection seems unnecessary. Does the PCA provide meaningful information?

The discussion should be rewrite taking in account that the discussion reviews the findings and puts them into the context of the overall research. In the current version, L. 279-309, 313-322, 326-346, 362-381 are more suitable for Introduction, whereas the rest part of discussion L. 309-312, 322-325, 347-360, 381-384 are repeating the results.

 Stylistic errors distort the meaning. For example, “the extract illuminated with mixed red/blue light”, “light quality“, “the blue light extract “, “The control white light extract“, “The highest phenolic and flavonoid content values were achieved after blue light illumination”, “Gram (-) and 4 Gram (+) bacterial strains”, “cultivated under different light quality”.

Author Response

We thank the Reviewer for the constructive comments that led to a considerable improvement of our manuscript.

Reviewer: The manuscript entitled “Blue light is an elicitor of the antioxidant, antimicrobial, and antitumor potential of extracts from the green microalga Coelastrella sp. BGV” by Georgieva et al. presents a study of different color light on the bioactive capacity of extracts from the Bulgarian strain of the green microalga Coelastrella sp. BGV. In this study the authors focused on impact of various color filtered LED lights on the content of phenolic antioxidants and the estimation of antioxidant, antibacterial, and cytotoxic activites of ethanol extracts from this green microalga.

After reading the entire manuscript, I believe that it requires a lot more work before it being considered acceptable for publication.

Comment 1:  The authors faced the problem of what to call the light. The light of different colors has received the following names: “three different LED light formulas, different LED light conditions, different light quality, different light condition, light variant, quality of light, differences of light variants, LED light sources, the light illumination conditions, influence of light, different light conditions were tested, influence of light quality”.

Response 1: For precision, everywhere throughout the revised manuscript, the light synonyms used to call ‘light of different colors’ were corrected and more accurately substituted to ‘light’ only.

Comment 2: “blue light extract, white light extract, red/blue extract” are inappropriate.

Response 2: These inappropriate terms were corrected throughout the revised manuscript.

Comment 3:  The manuscript lacks adequate discussion. The authors should realized that the purpose of the discussion section is to interpret and describe the significance of results obtained in relation to what was already known about the research problem under consideration, and to explain new understanding that emerged as a result of your research.

Response 3: We agree with the Reviewer's comment and rewrote the Discussion section (Lines 296-390)

The article was prepared carelessly.

Comment 4:  The language is poor enough that the authors should have a third party review it for style before resubmission.

Response 4: We worked on improving the language in the revised manuscript version.

Detailed comments:

Comment 5:  The title should be edited because (1), strictly speaking, in plant biology, elicitors are substances (see below) and (2) the light affected the microalgae, not the extracts. Maybe something like that: “Blue light enhances the antioxidant, antimicrobial and antitumor potential of the green microalgae Coelastrella sp. BGV”, or something else.

Response 5: We understand the Reviewer’s point and agree with the suggested changes in the title.

Comment 6:  Comments about “an elicitor”:

In plant biology, elicitors are generally the low-molecular-weight foreign molecules often associated with plant immune response by activating signal cascade. Elicitors are classified into two types, exogenous and endogenous, of elicitors, which can attach to receptor protein of the plant cell membrane.

An elicitor is a molecule that triggers the hypersensitivity response in the plant. Elicitors are very diverse molecules without any chemical similarity, except that they trigger the hypersensitivity response.

Response 6: Thanks for the careful information. We named the light ‘elicitor’ since light is an environmental factor that triggers signaling pathways that assure adequate responses. We understand the Reviewer’s point and agree with it. Changes are made throughout the revised manuscript and the term is not mentioned at all.

Comment 7:  L. 20. What extracts? Specify ethanol extracts.

Response 7: Ethanol extracts were specified. We occasionally used as terms ‘ethanol’ or ‘ethanolic’ extract/s, and in the revised manuscript version we decided to use only ‘ethanol’.

Comment 8: L. 27. Are you sure about the word “Unlikely”?

Response 8: We agree that “Unlikely” is inappropriately used and it should be replaced – “In contrast” was used instead (Line 26).

Comment 9: L. 37-38. 53, 56 Correct

Response 9: We apologize for the technical problem with some references’ citation, we worked on solving this issue.

Comment 10: I would recommend adding data about the effect of the light spectrum on the production of bioactive substances in microalgae to the introduction.

Response 10: We moved part of the information about the light spectrum on the production of bioactive substances in microalgae to the introduction (Lines 77-81).

Comment 11: Table 1. What are the values in the table? Mean and SD, n=? This information should be given.

Response 11: We agree that we missed this information and provided it in the revised manuscript in Table 1, as well as in the other Table or Figure legends where it was missing.

Comment 12: L. 104-108. Delete.

Response 12: We apologize for this repetition that remained unnoticed.

Comment 13: L. 109. “light quality”. Write more specifically.

Response 13: We tried to be more specific throughout the revised manuscript and used “light”.

Comment 14: L.116. Correlation analysis was not carried out.

Response 14: We agree to remove this sentence. Later in the 2.5. PCA section, the correlations between the parameters are analysed (Lines 264-295).

Comment 15: Table 3. Give the full names of bacteria and fungi.

Response 15: The full Latin names of the microorganisms in Table 3 are included.

Comment 16: Figures 2, 3 and 4. To compare the results (for better visualization and clarity), it is better to combine the columns on a single histogram. You can combine the results (columns) for 24 and 48 hours or for different lightings.

Response 16: Figures 2, 3 and 4 have been corrected according to the reviewer's recommendation. The columns are combined for different lightings.

Comment 17: Figures 2, 3 and 4. Number of experiments should be indicated in the legends to the figures and tables. So, data are presented as mean ± SD, n=?

Response 17: The legends to figures have been corrected according to Reviewer’s comment. n = 6.

Comment 18: Figure 4. What do the letters a, b and c mean? Specify.

Response 18: The letters are used in the Tables. We included information in the Table legends and in the method section 4.7. (Lines 526-543).

Comment 19: L.259-277. This subsection seems unnecessary. Does the PCA provide meaningful information?

Response 19: We believe that the PCA does provide meaningful information related to statistics and visualization. Principal component analysis is a powerful tool for visualizing differences between variants according to the whole experimental data. Each principal component (PC) describes part of the overall variance in the data (between variants). The first (PC1) explains the majority of the variance and the contribution of each following component diminishes with their number. Variants position according to the PC1/PC2 values. On the other hand, each experimental parameter determines each PC to different extend which is given by the length of each vector and its orientation in the PC1/PC2 coordinate system. Closer the vector is to one of the axises, higher its contribution to the respective PC. Thus, as outlined in the text, "overlapping arrows correspond to parameters with high positive correlation" and linearly opposing vectors point to highly negatively correlated variables, while orthogonally positioned variables characterize with no correlation between them. That way we used PCA instead of correlation analysis to present the same information more concisely in addition to summarizing all the findings in the separate sub-experiments. We worked on improving this section 2.5. (Lines 264-295). 

Comment 20: The discussion should be rewrite taking in account that the discussion reviews the findings and puts them into the context of the overall research. In the current version, L. 279-309, 313-322, 326-346, 362-381 are more suitable for Introduction, whereas the rest part of discussion L. 309-312, 322-325, 347-360, 381-384 are repeating the results.

Response 20: We agree with the Revier. In the revised manuscript, the Discussion section is rewritten. According to Comment 10, information about the effect of the light spectrum on the production of bioactive substances in microalgae was moved to the Introduction.

Comment 21:  Stylistic errors distort the meaning. For example, “the extract illuminated with mixed red/blue light”, “light quality“, “the blue light extract “, “The control white light extract“, “The highest phenolic and flavonoid content values were achieved after blue light illumination”, “Gram (-) and 4 Gram (+) bacterial strains”, “cultivated under different light quality”.

Response 21: We agree with the recommendation and here we list the changed phrases incorporated in the revised manuscript:

1) “the extract illuminated with mixed red/blue light” – “the C1 extract”

2) “light quality“ – “light”

3) “the blue light extract “ – “the C2 extract “

4) “The control white light extract“ – “The control C3 extract“

5) “The highest phenolic and flavonoid content values were achieved after blue light illumination” – “The highest phenolic and flavonoid content values were achieved in the C2 extract”

6) “Gram (-) and 4 Gram (+) bacterial strains” – “Gram-negative and 4 Gram-positive bacterial strains”

Reviewer 3 Report

Comments and Suggestions for Authors

I have read the manuscript and have several questions and recommendations.
1. Usually alcohol and alcohol-water mixtures are used for degreasing algae. Polyphenol fractions are obtained using, for example, ethyl acetate for extraction of Padina boergesenii, modern green solvents NADES from Arctic Fucus vesiculosus, etc. Please justify the choice of extractant.
2. Please indicate which positive and negative controls were used for antitumor activity.
3. Discuss the data for PCA in more detail.
4. Provide the calculation formula for total antioxidant activity.
5. In the "Materials and Methods" section, indicate the methods for determining pigments and the formulas of their calculation.
6. In Tables 1 and 2, reduce the number of significant digits after the decimal point.
7. What is the content of fatty acids in the extracts? Does light affect the change in fatty acids? Compare with literature data.

Author Response

We thank the Reviewer for the constructive comments that led to a considerable improvement of our manuscript.

Reviewer: I have read the manuscript and have several questions and recommendations.
Comments 1:  Usually alcohol and alcohol-water mixtures are used for degreasing algae. Polyphenol fractions are obtained using, for example, ethyl acetate for extraction of Padina boergesenii, modern green solvents NADES from Arctic Fucus vesiculosus, etc. Please justify the choice of extractant.

Response 1: Ethanol has been widely applied as a solvent in the last few decades and is classified as an environmentally preferable green solvent because it is produced by fermentation of renewable sources (sugars, starches, etc.). Ethanol is a green, safe, and environmentally friendly solvent that has GRAS (Generally Recognized As Safe) status by FDA (United States Food and Drug Administration) and is approved by EFSA (European Food Safety Authority). Therefore, ethanol extraction is a technology safe for human health and the environment. Ethanol is a suitable solvent for the extraction of natural compounds from microalgae that belong to a wide range of chemical classes, including phenols, pigments (carotenoids and chlorophylls), fatty acids, polysaccharides, vitamins and others. In the present study, we are focused on the evaluation of ethanolic extracts from Coelastrella sp. BGV, (obtained under three different lights) in terms of yield, phenolic content, antioxidant and potential antitumor activity in in vitro experiments. The studied crude ethanolic extracts are a mixture of bioactive compounds, and further analyzes will be needed to identify the bioactive components in the extracts and their possible synergistic action.

In the revised manuscript, in section 2.2. we included Table S1, where we compare the content of phenolic antioxidants in ethanol and water solvents (Lines 105-106). The data showed that the ethanol is enriched in phenolics and therefore is suitable to further investigation. A justification of the choice of extract in the case of Coelastrella sp. BGV is incorporated in the Discussion section (Lines 319-322; 369-372).

Comments 2:  Please indicate which positive and negative controls were used for antitumor activity.

Response 2: In the present pilot study we used two tumor cell lines as a model system of common human malignancies – HeLa (cervical carcinoma) and HepG-2 (hepatocellular carcinoma). The "gold" standard in cytotoxicity tests is the use of fibroblasts of human or animal origin as an analogue of healthy cells. Such "normal" standard are BALB/3T3 mouse fibroblast cells. The antiproliferative activity was compared in two aspects – against cells of the corresponding type (tumor and non-tumor) untreated with ethanol extracts (negative control), as well as between tumor and normal cells. The study allows to assess the antitumor effect of the ethanol extracts and the presence of selective action against tumor cells.

In the Discussion section, we included comments on the controls for antitumor activity (Lines 367-377).

Comments 3:  Discuss the data for PCA in more detail.

Response 3: In the revised manuscript, a more detailed discussion of the PCA data is included (Lines 264-295).

Comments 4:  Provide the calculation formula for total antioxidant activity.

Response 4: We provided the formula in the Method section 4.3 (Lines 435-437). As well, we included the formula modified for phenolics and flavonoids (Lines 431-434).

Comments 5:  In the "Materials and Methods" section, indicate the methods for determining pigments and the formulas of their calculation.

Response 5: We apologize for missing this method. We included it in the revised manuscript in section 4.3. (Lines 409-423).

Comments 6:  In Tables 1 and 2, reduce the number of significant digits after the decimal point.

Response 6: We followed this recommendation in the revised manuscript.

Comments 7:  What is the content of fatty acids in the extracts? Does light affect the change in fatty acids? Compare with literature data.

Response 7: We did not measure the content of fatty acids in the studied ethanol extracts. We discuss the content of Coelastrella sp. BGV lipids in ethanol extracts in the revised Discussion section (Lines 308-313).

In the literature, Amaro et al. [54] investigated whether monochromatic blue (B) and red (R) LEDs and two dichromatic combinations of blue and red (BR (40:60) and BR (50:50)) and fluorescent light (FL) as a control exerted an influence on the production of antioxidant compounds by Scenedesmus obliquus M2-1. The results showed that B induced higher biomass productivity and higher fatty acid content compared to other tested LED conditions (Lines 382-384).

Round 2

Reviewer 2 Report

Comments and Suggestions for Authors

In the revised version the authors took into consideration all comments and remarks. I recommend to accept your manuscript for publication in Plants.

Author Response

Reviewer: In the revised version the authors took into consideration all comments and remarks. I recommend to accept your manuscript for publication in Plants.

Response: Thank you again for the constructive comments and for contributing to the substantial improvement of the paper.

Reviewer 3 Report

Comments and Suggestions for Authors

I have read the revised manuscript. The authors have not fully answered my questions.

1. What do "a", "b" and "c" mean in Tables 1, 2, 4 and 5.

2. In Tables 3, 4 and 5, indicate the standard deviation. Or were the data not normally distributed?

3. Alcohol and alcohol-water mixtures are usually used to degrease algae. Polyphenol fractions are obtained using, for example, ethyl acetate for the extraction of Padina boergesenii, modern green solvents NADES from Arctic Fucus vesiculosus, etc. Justify the choice of extractant. Supplement the introduction with literary data on various types of modern green solvents. The use of alcohol is not new.

Author Response

Reviewer: I have read the revised manuscript. The authors have not fully answered my questions.

Comment 1: What do "a", "b" and "c" mean in Tables 1, 2, 4 and 5.

Response 1: We tried to optimize the Tables’ legends to avoid confusion.

The different letters mean statistical differences between the extract variants. For clarity, we re-wrote the Table legends:

Table 1. Pigment content of Coelastrella sp. BGV cultivated at different illumination conditions (C1 – red/blue light; C2 – blue light; C3 – white light). Data are presented as mean ± SD (n ≥ 6). A one‒way ANOVA (Holm‒Sidak test) was applied to determine statistical differences between the extract variants, which are denoted by different letters for each pigment (p ≤ 0.05). (Lines 99-102)

Table 2. Yield, total phenolic and flavonoid content, and TAA of ethanol extracts of Coelastrella sp. BGV cultivated at different illumination conditions (C1 – red/blue light; C2 – blue light; C3 – white light). Data are presented as mean ± SD (n ≥ 6). A one‒way ANOVA (Holm‒Sidak test) was applied to determine statistical differences between the extract variants, which are denoted by different letters for each parameter (p ≤ 0.05). (Lines 111-115)

Table 3. Antimicrobial activity of ethanol extracts of Coelastrella sp. BGV cultivated at different illumination conditions (C1 – red/blue light; C2 – blue light; C3 – white light). Data are presented as mean ± SD (n ≥ 3). The zone of inhibition showed a bactericidal effect. *Ciprofloxacin (for the bacteria) and nystatin (for the fungus) were positive antibiotic controls. (Lines 131-134)

We combined Tables 4 and 5 into a new Table 4.

Table 4. IC50* and Selectivity index (SI)** values of ethanol extracts of Coelastrella sp. BGV cultivated at different illumination conditions (C1 – red/blue light; C2 – blue light; C3 – white light).

* IC50 (µg/mL) values of the microalgal ethanol extracts on cancer (HeLa; HepG2) and normal (BALB/3T3) cell lines, determined at 24th and 48th. Data are presented as mean ± SD (n = 6). A one‒way ANOVA (Holm‒Sidak test) was applied to determine statistical differences between the extract variants, which are denoted by different letters for each time point (p ≤ 0.05).

** The SI value (in parentheses, in bold) was calculated from the IC50 ratio of BALB/3T3 cells versus IC50 of cancer cells HeLa or HepG2, respectively. The selectivity score was separated into three categories: SI values ≤ 1 – non-selective action; 1 < SI values < 5 – moderate selectivity; SI values > 5 – selective action [38].

The information about statistics is described also in the section Methods (Lines 533-550).

Comment 2: In Tables 3, 4 and 5, indicate the standard deviation. Or were the data not normally distributed?

Response 2: We worked on Tables’ optimization.

In Table 3 and 4 we indicated the standard deviation.

For clarity, we decided to combine Tables 4 and 5 into a new Table 4 (Lines 203-218). The SI value represents ratios between the average IC50 values of the ethanol extracts on the normal and the cancer line, and we showed the highlighted SI value, as the statistics is reflected by the ratios.

Comment 3: Alcohol and alcohol-water mixtures are usually used to degrease algae. Polyphenol fractions are obtained using, for example, ethyl acetate for the extraction of Padina boergesenii, modern green solvents NADES from Arctic Fucus vesiculosus, etc. Justify the choice of extractant. Supplement the introduction with literary data on various types of modern green solvents. The use of alcohol is not new.

Response 3: Thanks for the useful recommendation.

In the previous manuscript revision, to justify the choice of ethanol, we included in Results data that ethanol extracts are enriched in phenolics compared to water extracts, and therefore, we continued the work with the ethanol extracts (Lines 104-105). In the Discussion we pointed to the high antimicrobial and antiproliferative potential of Coelastrella sp. BGV ethanol extracts (Lines 351-376).

In this revised manuscript version, we included an improved justification of the choice of solvent in the Discussion (Lines 320-330) and Conclusion section (Lines 559-562). In our study, we emphasize on the importance of cultivation conditions for optimizing microalgal productivity, namely illumination during microalgal cultivation. We did not investigate the effect of solvents and novel alternatives, however, as mentioned above, we included justification which was missing indeed.

Round 3

Reviewer 3 Report

Comments and Suggestions for Authors

No questions